# Changes in Management Trends in 100 Global Companies before and after COVID-19: A Topic Modeling Approach

**Hyeonjeong Park [1], Taewoo Kim [1] and Keuntae Cho [2,*]**

[1] Graduate School of Management of Technology, Sungkyunkwan University, Suwon 16419, Republic of Korea; hjppulip94@skku.edu (H.P.); taewoo2040@skku.edu (T.K.)
[2] Department of System Management Engineering & Graduate School of Management of Technology, Sungkyunkwan University, Suwon 16419, Republic of Korea
[*] Correspondence: ktcho@skku.edu; Tel.: +82-031-290-7602

**Abstract:** Amid the global economic crisis due to COVID-19, consumer interest in CSR reports of global corporations has surged. CEO messages within these reports are crucial during such crises. This study aims to understand CEOs' perceptions of key CSR issues and corporate strategies during global crises. Utilizing topic modeling, CEO messages from Fortune Global's top 100 companies are analyzed before and after COVID-19. Compared to previous periods, social and environmental issues like climate change are more prominent. Key strategies include sustainable management development, risk management, and competitive advantage. This study offers insights into the importance of CSR reporting as a communication tool for managing stakeholders during crises. With COVID-19's global impact, understanding changes in global companies and comparing pre-crisis conditions contributes significantly to the literature.

**Keywords:** corporate social responsibility (CSR); corporate social responsibility (CSR) report; CEO message; sustainability; topic modeling; COVID-19

## 1. Introduction

During the coronavirus disease (COVID-19) pandemic, social distancing was recommended for citizens worldwide. Quarantine measures were implemented, and movement restrictions and non-face-to-face situations were advised for all essential activities. Countries and cities were sealed off, and social and economic activities as well as epidemic control measures due to quarantine led to a reduction in labor supply, income losses due to deteriorating economic forecasts, and decreased business investment. This affected businesses worldwide in terms of both supply and demand [1]. It is therefore necessary to address the impact of COVID-19 on global companies around the world [2].

The unprecedented scale of the crisis has compelled companies to acknowledge the evolving environment promptly and adapt by restructuring their business models. This restructuring encompasses strategies and operations with a focus on stakeholders. Failure to do so may result in losses for various domestic and international stakeholders, including customers, employees, investors, and others [3–6].

Various stakeholders place greater importance on the ethical aspects of consumption. There is growing public interest in and demand for corporate social responsibility (CSR) activities, which involve the production of goods and services needed by society as well as corporate social and environmental responsibility [4,5,7,8]. CSR serves as a business strategy management tool for handling a company's diverse stakeholders, and significantly affects its operations [9]. Consequently, corporate managers are increasingly preparing non-financial reports (e.g., CSR and sustainability reports) to maintain stakeholders' trust, protect stock prices, and enhance corporate reputation. These reports include sustainability information on CSR activities, such as employee welfare, community involvement, and environmental initiatives, with increasing frequency [10,11].

Non-financial reporting serves as a corporate strategy capable of enhancing a company's reputation and generating diverse business revenue. It functions as a communication tool between a company and various stakeholders, particularly for conveying CSR [12]. Additionally, it has proven effective in addressing negative news and financial losses during crises [13–15]. Consequently, it has become a particularly valuable tool for global companies that have established trust not only with domestic but also international stakeholders, especially in the context of international crises such as COVID-19 [16,17].

However, previous studies have focused primarily on internal corporate crises, with limited research on CSR strategies and communication during international external crises [13,18,19]. This study aims to compare the perspectives of CEOs from Fortune Global's top 100 companies as of 2022 and the business situation before and after COVID-19. Three research questions are posed: (1) Do Fortune Global's top 100 companies utilize CSR reports? (2) Are there differences in keywords in CEO messages before and after COVID-19? (3) What CSR issues do CEOs perceive, and how have they changed before and after COVID-19? Research question (1) aids in exploring CSR issues published annually by these companies and provides insight into communication focused on CEO messages. Questions (2) and (3) further delve into comparing results obtained from periods before and after COVID-19 to examine changes in business trends. Therefore, this study aims to collect CEO messages from CSR reports of global companies, analyze them using keyword and topic modeling, and understand changes in corporate management trends before and after COVID-19.

## 2. Literature Review

### 2.1. Corporate Social Responsibility (CSR)

CSR is defined differently by scholars, institutions, and societies [20]. In the early 1960s, it was characterized as the decisions and actions of businesspeople, encompassing not only the economic and legal obligations that society has to corporations but also the ethical and discretionary expectations [21,22]. Currently, corporate social responsibility is defined as the concept of acknowledging a corporation's ethical responsibility to society and integrating it into interactions with customers, suppliers, and other stakeholders [5]. It is also defined as a concept that emphasizes that companies have a moral responsibility toward society and should integrate it into their interactions with stakeholders [5]. Therefore, this study defines CSR as a means for companies to enhance their social welfare and strengthen their relationships with different stakeholders [23].

Prior studies indicate that CSR activities have garnered significant interest in recent years, not only among academics and researchers but also among the general public [24]. Communication through various CSR activities is the most effective way for companies to demonstrate their commitment to fulfilling their responsibilities and respond to the demands of key stakeholders and active shareholders. Consequently, CSR communication has emerged as a crucial aspect of crisis management research [14,25].

The objective of CSR communication is to enhance consumers' knowledge, awareness, trust, and engagement in CSR and to shape their perceptions of a company's reputation. Specific content and media channels, such as reporting, are employed to cultivate trust-based relationships with customers and shareholders, as well as to inform stakeholders about CSR strategies and practices [26,27].

CSR reporting serves as a CSR activity and a communication tool between a company and its stakeholders. Types of CSR reports are divided into financial and non-financial reports, with non-financial reports including corporate social responsibility (CSR) reports, sustainability reports (SR), ESG reports, and integrated reports (IR), among others [27]. However, in 1999, social and environmental reports were integrated into sustainability reports. According to previous studies, sustainability reports are still published based on guidelines such as the Global Reporting Initiative (GRI) and Sustainable Development Goals (SDGs), and are commonly referred to alongside the term CSR (corporate social responsibility) [28,29].

CSR reports contribute to the systematic management of corporate social responsibility activities, helping to identify future risks and crises and ensure the sustainable development and competitive advantage of the company [30]. It has also been demonstrated that most socially responsible companies act more responsibly when preparing their CSR reports [31]. Therefore, this study concentrates on CSR reports that incorporate non-financial information as a means of corporate communication to enhance a company's reputation and engage various stakeholders.

### 2.2. Prior Research on COVID-19 and CEO Messages

The first outbreak of COVID-19 in Wuhan, Hubei Province, China, in December 2019 led to the rapid spread of the disease worldwide, including in Europe and the United States [32]. In response, the World Health Organization (WHO) declared a "public health emergency of international concern" on 30 January 2020, to address the global spread of COVID-19. In March, the WHO upgraded the situation to a "pandemic", highlighting the global impact of COVID-19. As a new infectious respiratory disease, COVID-19 rapidly spread worldwide owing to its strong contagious nature and significant ripple effects. It began to pose persistent economic, social, and environmental problems after severely impacting the global economy [2].

The COVID-19 pandemic is the most recent instance of an externally generated crisis—a severe global crisis that has resulted in economic disruptions—with effects such as income inequality, racism, and climate change, altering daily life and adversely affecting business processes [33–35]. This underscores the role that companies must play in society and heightens the demand for CSR activities [1,5].

Moreover, companies are integral to the SDGs known as the 2030 Sustainable Development Agenda, which addresses crises and sustainability issues. During the COVID-19 period, many global companies recognized the changing environment and expanded their CSR strategies accordingly, swiftly transforming their business models [1,5]. Consequently, CSR activities are expected to continue evolving, and further examination of CSR-related topics is warranted to understand the impact of COVID-19 on businesses [36].

Reorienting and developing business processes toward CSR and addressing the challenges posed by COVID-19 necessitate decision-making at the board and CEO levels [2]. This underscores the importance of corporate crisis-response strategies that dictate what companies communicate and undertake during a crisis [37]. A CEO's message in a CSR report has emerged as the most visible manifestation of sustainability leadership.

In a CSR report, the will of the CEO and the company profile are initially highlighted, setting the overall tone of the report [30,38]. The CEO serves as the voice of the company and conveys substantial information that affects its operations. Through this report, a company can positively influence stakeholders and enhance its image [39]. Consequently, the relevance of the COVID-19 pandemic to CSR can be perceived differently from the CEO's perspective compared with that of various stakeholders [36].

It has been observed that the majority of studies conducted during COVID-19 focused on the perspectives of employees and customers in relation to reporting and CSR communication [40]. Specifically, financial reports were used to examine their correlation with fraud in financial statements, changes in profitability, and other factors [36]. However, there is a lack of analysis that accurately distinguishes between CEO messages presented in the reports and the time points before and after COVID-19 concerning the flow of corporate trends [36]. Furthermore, although COVID-19 is an international crisis, there is insufficient research on CSR strategies for global companies in various industry sectors, with results drawn from specific industries or countries [18].

Therefore, this study utilizes keywords and modeling analysis to investigate changes in management trends based on the messages of CEOs of global firms before and after COVID-19. The aim is to understand their strategies for responding to crises. It is particularly meaningful to compare the analysis results from the perspectives of CEOs before and after the COVID-19 crisis.

## 3. Materials and Methods

### 3.1. Data Collection

This study targeted the top 100 companies listed in the Fortune Global 500, selected through purposive sampling. Purposive sampling refers to the researcher's judgment and subjective construction of the sample [41]. While random sampling is another option, purposive sampling was chosen because higher-ranked companies are known to engage more actively in CSR activities and utilize CSR reports more extensively. Using random sampling could lead to the exclusion of major companies with significant CSR reports, thus limiting the study's contribution. Moreover, many previous studies have focused on CSR research based on the top 100 companies, considering it a reliable sample size [42]. Therefore, this study utilized Fortune Global data to explore the broad impact of COVID-19 on multinational corporations and to analyze differences in corporate management strategies before and after COVID-19 from the perspective of CEOs of top global companies by collecting and analyzing data from two distinct periods.

The data collection period was divided into pre-COVID-19, covering the years from 2017 to 2019, and after COVID-19, covering the years from 2020 to 2022. CEO messages extracted from CSR reports were collected by visiting each company's website. Only messages from the CEO message section of CSR reports from 2017 to 2022 were excerpted. Additionally, 30 out of the total 100 companies were excluded from the study as their reports for six consecutive years were unavailable or did not contain CEO messages. Therefore, text data from a total of 70 companies were converted into TXT format for topic modeling analysis. This study ultimately selected 420 CEO message sentences, representing each year, for analysis. Through this data analysis, keywords and topics emerging in each period were extracted, and the researcher manually named the topics based on the characteristics of the extracted keywords and topics.

### 3.2. Research Procedures and Analysis

In this study, the R programming language was employed to refine the data and conduct keyword and topic-modeling analyses for the periods before and after COVID-19. The data analysis program used for the Latent Dirichlet Allocation (LDA) analysis was R version 4.0.2. It is commonly used for data mining, data visualization, artificial intelligence, and big data. The program has additional features called packages; a Natural Language Processing-based TM package was utilized in this study [43]. We also utilized the LDA library belonging to the "topicmodels" package. To recognize the frequency of keywords and differences intuitively and rapidly before and after COVID-19, the "LDA vis" package was employed for visualization. A flowchart of the study is shown in Figure 1.

#### 3.2.1. Data Preprocessing

To conduct keyword analysis (topic analysis) of the collected TXT files, the Natural Language Toolkit provided by the R program was utilized, employing the corpus's non-use term dictionary and applying several conditions, including the exclusion of non-use words [43]. Initially, capitalized words such as "Business", "Energy", "ESG", and "Carbon" were converted to lowercase to standardize their appearance with lowercase words such as "business", "energy", "esg", and "carbon". Subsequently, a preprocessing step was implemented to remove special characters and other unnecessary information. Unused terms consisted of meaningless words such as articles ("a", "an", "on", "of", "because"), prepositions, interrogatives, conjunctions, auxiliary verbs ("will"), and adverbs ("also", "even", "also"), along with words and numbers that hold no analytical significance. After eliminating these meaningless words, only meaningful sentences were retained and analyzed [43]. Additionally, names representing companies, such as Huawei, Samsung, Volkswagen, and Walmart, were excluded because they did not contribute meaningfully to the analysis.

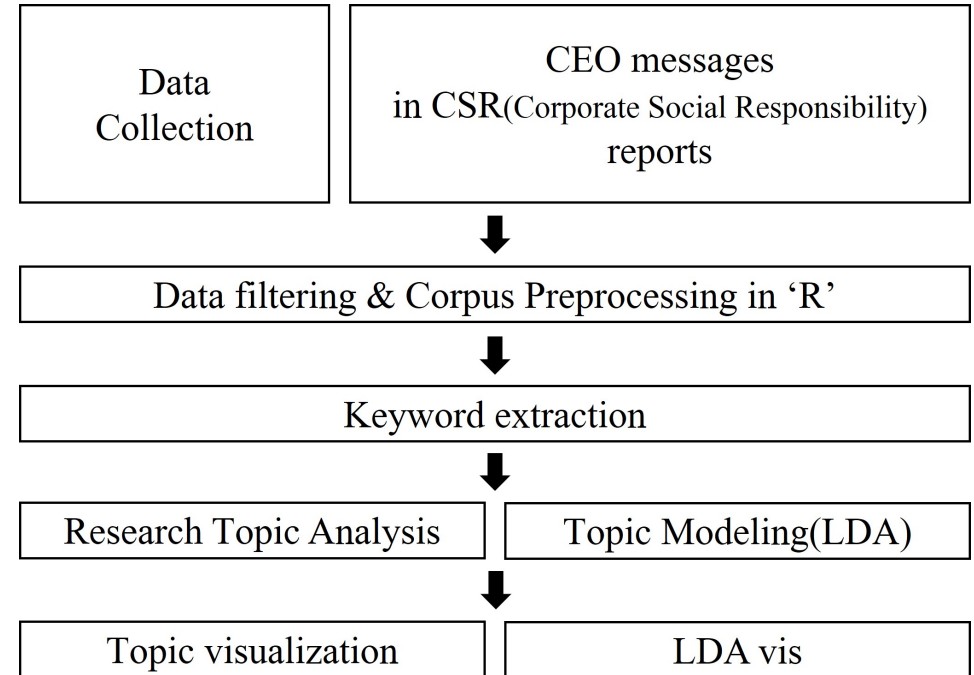

**Figure 1.** Research flowchart.

3.2.2. Keyword Analysis

The top 30 most frequently appearing words in CEO messages were ranked and analyzed for each period. Analyzing theme keywords and related keywords in order of frequency of occurrence is useful to intuitively comprehend the extensive content of the collected data [44]. Related keywords can be considered words that reflect interest and concern for theme keywords. Therefore, through a comparison before and after COVID-19, this study aimed to confirm, through keyword frequency analysis, the main themes in which firms are interested and the changes that have occurred due to COVID-19.

3.2.3. Topic-Modeling Analysis

A topic model, specifically LDA, is a statistical model used to discover themes in a set of documents. It is a text-mining method designed to reveal hidden semantic structures within a text body. The results of LDA are particularly interpretable, making it a method for extracting various meaningful main topics from a large amount of textual data [45]. Widely employed as a research methodology, it is used to extract major research topics, primarily by analyzing texts such as article abstracts and news articles to identify research trends over time [46]. This methodology does not merely classify themes; instead, it identifies the keywords contained in the themes, which are then utilized to interpret and define the themes in question. The LDA results were used to name themes based on the researcher's direct observation and judgment.

In topic modeling, the derivation of a significant number of topics is crucial. Many researchers have determined the number of topics by evaluating various model fit indices, including the perplexity and interpretability of the topics [46]. The x-axis represents the number of topics, the y-axis represents the perplexity value, and the perplexity is determined by varying the number of topics from 3 to 15. After determining the number of topics, the interval in which the perplexity value minimizes the difference is chosen. Therefore, this study aimed to determine the appropriate number of topics using perplexity. The results of determining the perplexity value using keywords extracted from the CEO messages revealed that the perplexity value was in the 5–6 interval in the first semester and in the 6–7 interval in the second semester, representing the point at which the value was minimized. Therefore, in this study, six topics were identified before COVID-19 and

seven after the COVID-19 outbreak. Figures 2 and 3 show the diagnostic values used to determine the number of topics.

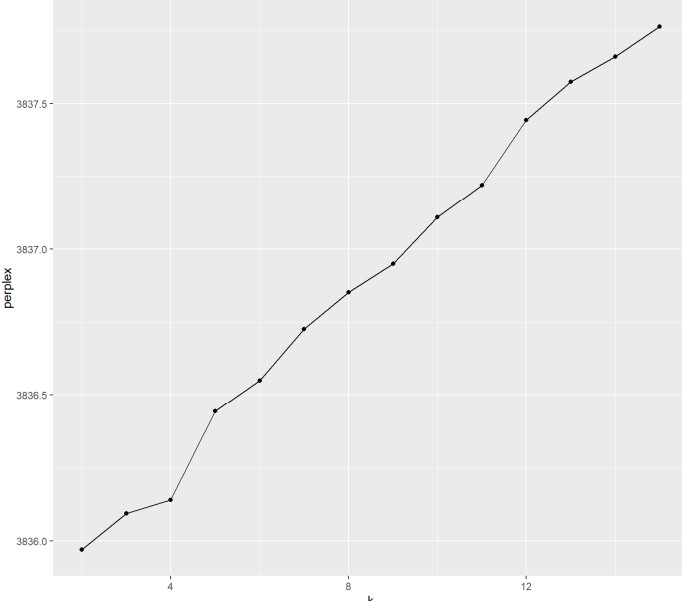

**Figure 2.** Before COVID-19 data.

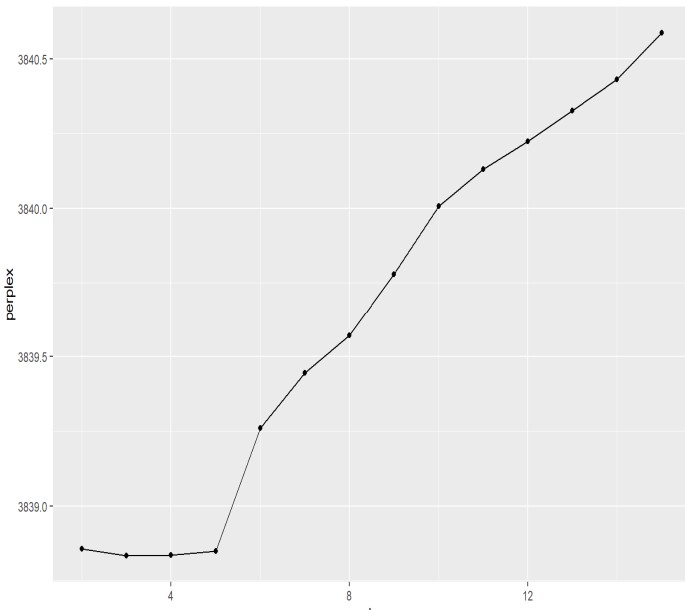

**Figure 3.** After COVID-19 data.

The visualization of the results of topic modeling in LDA comprises a topic map that spatially displays the distance between circles representing a topic. In addition, a list of the top 30 keywords that constitute the topic is presented in the form of a bar graph. The topic circles in the topic map do not overlap, and the greater the distance between them, the higher the discriminant validity of the topic.

## 4. Results

### 4.1. Keyword Analysis Results

This study analyzed keyword frequency for CEO messages in the CSR and SR of the Fortune Global 100 largest companies before (2017–2019) and after (2020–2022) COVID-19,

through the preprocessing step. The 30 representative keywords are presented in order of frequency and the results derived, as shown in Table 1, before COVID-19 and Table 2 after COVID-19.

**Table 1.** Organizing the top "before COVID-19" words.

| Time | Year | Top 30 Frequent Words |
|---|---|---|
| Before COVID-19 | 2017 | business, customer, company, new, growth, service, development, management, value, group, world, market, time, opportunity, energy, global, industry, future, increase, health, cost, result, asset, continue, technology, work, people, network, share, sustainability |
| | 2018 | business, new, company, growth, development, service, group, world, management, global, value, future, people, time, market, industry, energy, work, health, employee, continue, technology, network, opportunity, increase, strategy, share, quality, importance, asset |
| | 2019 | business, company, customer, new, development, growth, service, group, management, world, value, future, work, health, employee, continue, technology, market, global, time, people, industry, energy, environment, support, risk, quality, share, system, asset |

**Table 2.** Organizing the top "after COVID-19" words.

| Time | Year | Top 30 Frequent Words |
|---|---|---|
| After COVID-19 | 2020 | business, company, customer, new, development, growth, service, group, global, world, management, market, energy, people, health, value, work, continue, time, industry, sustainability, future, network, employee, risk, effort, environment, support, community, digital |
| | 2021 | business, customer, new, company, development, service, group, global, work, pandemic, people, world, energy, management, employee, value, carbon, support, society, continue, community, market, time, COVID, digital, effort, environment, industry, health |
| | 2022 | business, company, new, customer, service, development, growth, world, people, group, global, work, management, energy, community, industry, market, value, health, employee, time, technology, future, continue, digital, sustainability, share, effort, system, support |

First, irrespective of the status of COVID-19, the top five keywords common to the first and second semesters are "business", "company", "new", "service", and "development". The development of new products and services is considered an essential element of a firm's outcome [47]. Therefore, these five common keywords represent the most important perspectives that a company's CEO focuses on.

Comparing before and after COVID-19, the common keywords for the period from 2017 to 2019 are "technology" and "share", indicating that the content related to technological innovation and the culture of sharing was prominent during that time. In contrast, the common keywords for the period from 2020 to 2022 are "effect", "community", and "digital", suggesting active discussions about efforts to overcome the COVID-19 situation, community-centered initiatives, and digitalization.

*4.2. Topic-Modeling Analysis Results*

In this study, the R program was used to extract key topics and conduct a topic-modeling analysis of CEO messages from Fortune Global 100 companies at points in time before and after the COVID-19 pandemic.

The first topic derived in Table 3, before COVID-19, consists of related words: growth, business, work, risk, sustainability, platform, mobility, and improvement. This topic comprises the content related to risk management. Risk management is a core element of the strategic activities of all organizations [48]. Common risk factors related to sustainability in various industrial sectors, such as logistics and transportation, include greenhouse gas emissions, natural disasters, energy consumption, and environmental damage [49]. Consequently, ongoing disruptions in social, political, and economic environments; competition among firms; and technological development have urged firms to establish robust risk management systems [50].

**Table 3.** Major topics before COVID-19.

| Group | The Top 30 "before COVID-19" Words | Topic |
|:---:|---|---|
| 1 | growth, market, business, time, work, risk, sustainability, system, efficiency, network, board, earnings, provide, resource, areas, internal, key, launch, carbon, chain, focus, platform, mobility, climate, improve, general, united, infrastructure, models, return | Risk Management |
| 2 | business, development, strategy, industry, health, opportunity, increase, global, environment, community, plan, care, reform, deliver, healthcare, improve, revenue, oil, post, organization, control, report, serve, country, responsibility, retail, safety, rate, focus, expect | Market Expansion |
| 3 | technology, support, sales, capability, significant, transformation, process, addition, cloud, commercial, cost, application, increasing, including, real, interest, reduce, largest, period, order, 5G, ability, price, trust, volume, tax, target, proud, dividend, job | Operational Efficiency |
| 4 | customer, future, employee, innovation, people, result, solution, profit, fiscal, project, insurance, partners, portfolio, policy, cost, success, team, scale, enhance, including, benefit, culture, emission, record, stock, current, improved, local, promote, times | Customer Centric Innovation |
| 5 | company, group, world, potential, asset, share, quality, corporate, progress, digital, change, capital, experience, increased, data, society, commitment, core, supply, past, initiative, model, position, mobile, development, approach, drive, generation, foundation, lead | Digital Integration |
| 6 | service, management, energy, economy, importance, strong, performance, create, vehicle, effort, gas, end, goal, power, bank, cash, store, brand, intelligence, global, demand, vision, network, continue, issue, achieve, place, consumer, electric, class | Energy Innovation |

The second topic consists of related words such as development, strategy, opportunity, global, community, healthcare, revenue, oil, retail, and expectations. This topic comprises content related to market expansion. Global companies incur higher fixed costs for market expansion through new openings in order to pursue new opportunities [51–53]. The frequency of international expansion in a given year also determines the level of internationalization in the following year, which is an important factor in increasing firm level [54].

The third topic consists of terms such as technology, capability, process, cloud, commercial, application, 5G, price, trust, tax, dividend, and others. This topic comprises the content related to operational efficiency. Companies have been migrating software-based applications to the cloud (the process of moving applications, data, infrastructure, security, and other objects to a cloud-computing environment) or distributing them to deliver new services and value more quickly, improve operational efficiency, and increase revenue. Thus, the Fourth Industrial Revolution has seen an increased level of interest in the need for innovative information technology (IT) infrastructure through automation, as companies worldwide take advantage of IT applications to leverage advanced technologies [55].

The fourth topic consists of related words such as customers, employees, people, solutions, profits, policies, benefits, and culture. This topic comprises content related to customer-centered innovation and culture. The pandemic has highlighted the ethical aspects of consumer decision-making, leading to a shift towards more responsible and socially conscious consumption. This change is likely to be reflected in corporate operations [4]. As the structure generates revenue and profit by selling to customers, it has been proposed that companies must focus on developing core competencies that help create sustainable customer satisfaction [56]. Therefore, it is evident that stakeholders have a significant interest in customer-centric cultures.

The fifth topic consists of related words such as company, potential, corporate, progress, digital, change, experience, data, supply, mobile, and lead. This topic comprises content related to digital integration. During the Fourth Industrial Revolution, the development of IT saw a surge in interest that demanded a new term: digitalization [57]. However, despite the diversity of technological innovations and methods, actual digital transformation has taken much longer and faced more difficulties than digital business development companies, which is expected to be owing to resistance to change and other reasons [58]. This indicates that companies focus on digital marketing across platform and business domains.

The sixth topic consists of terms such as service, management, energy, economy, vehicles, gas, power, demand, vision, network, and electricity. This topic comprises content related to energy innovation. The increase in greenhouse gas emissions, a significant concern for economic growth, has led to changes in sustainable environmental regulations through energy efficiency and other low-carbon technologies [59–62]. Additionally, smart grids were introduced during this period, indicating that content related to smart devices and renewable energy sources were major issues.

After COVID-19, the first topic derived in Table 4 shows that growth, economy, fiscal, serve, diversity, interest, united, individual, post, contribute, and so on are composed of related words. This is a new value creation topic, indicating that the content is related to the creation of new value. Value creation is at the core of business model research, enabling new business practices in crises of unprecedented scale and promoting economic value creation worldwide, led by companies such as Amazon, Facebook, and Google [63]. Therefore, it can be viewed as a strategy by which companies contribute to corporate outcomes through new value creation during the COVID-19 pandemic [63].

**Table 4.** Major topics after COVID-19.

| Group | The Top 30 "before COVID-19" Words | Topic |
|-------|-------------------------------------|-------|
| 1 | growth, economy, quality, opportunity, vehicle, plan, improve, fiscal, government, home, inclusion, serve, diversity, key, success, trust, interest, potential, stable, control, united, enable, individual, speed, market, post, compared, increasingly, aim, contribute | Value Creation |
| 2 | global, technology, pandemic, COVID-19, progress, transformation, supply, performance, power, create, environment, country, past, insurance, care, safety, intelligence, generation, key, improved, model, bank, local, reach, announce, leverage, dividend, responsibility, associates, lower | Environmental Sustainability |
| 3 | industry, network, effort, including, strong, corporate, solution, risk, 5G, commitment, organization, security, launch, role, information, result, strategy, employee, improving, related, real, commercial, foundation, advantage, class, segment, structure, poverty, report, confidence | CSR and Leadership |
| 4 | work, management, energy, market, health, sustainability, share, strategy, issue, employee, emission, initiative, end, social, chain, significant, goal, partners, climate, addition, platform, revenue, life, program, profit, cloud, infrastructure, national, equity, esg | Climate Change Initiative |
| 5 | world, system, focus, sales, capability, board, capital, areas, resource, policy, drive, market, team, environment, benefit, experience, access, purpose, approach, measures, efficiency, principle, ability, finance, natural, income, action, price, neutral, lead | Ethical and Compliance Policies |
| 6 | business, group, continue, time, innovation, future, increase, gas, importance, data, carbon, cost, project, provide, internal, governance, esg, reform, cash, oil, enterprise, loan, scale, consumer, earnings, challenge, place, public, members, reduce | ESG Management |
| 7 | company, customer, development, service, people, community, digital, support, asset, society, change, increased, healthcare, core, achieve, culture, smart, strategy, position, post, current, target, ai, risk, open, representing, built, medical, state, oriented | Digital Transformation |

The second topic consists of related words, such as global, technology, pandemic, COVID-19, transformation, supply, environment, insurance, and safety. It comprises content related to environmental sustainability. Owing to the COVID-19 pandemic, the importance of corporate environmental sustainability strategies has skyrocketed as a sustainable development goal to be achieved now and in the future [64]. Therefore, it can be seen as a strategy through which companies focus on value creation, not only in the internal environment but also on external environmental factors.

The third topic consists of related words such as industry, network, corporate, solution, risk, 5G, organization, role, strategy, and report. It comprises content related to CSR and leadership. Leadership is one of the core elements in developing and implementing CSR strategies; from a CSR perspective, leadership improves overall corporate performance and is considered a core element of business ethics [65–69]. In particular, the support of leaders is critical to the success of CSR outcomes [70,71], as senior management, such as

the CEO, are closely related to the successful establishment and implementation of CSR policies. Therefore, this can be viewed as a strategy by which companies seek to improve their corporate outcomes through CSR activities.

The fourth topic consists of the following related words: management, energy, health, sustainability, share, emissions, initiative, climate, platform, life, profit, cloud and esg. It can be seen that this comprises content related to climate change initiatives. The COVID-19 pandemic has brought environmental issues to the forefront, with increased attention on climate change and global warming. As a result, interest in carbon capture and utilization has also risen simultaneously [70]. Particularly, within corporations, the $CO_2$ value chain is considered to be a significant issue as it transcends mere emission reduction activities and becomes integral to business operations [70]. Hence, it can be observed that global companies are preparing for sustainable development with a heightened focus on carbon neutrality after COVID-19.

The fifth topic consists of the following related words: world, system, sales, board, policy, experience, access, purpose, efficiency, and principles. It can be seen that this comprises content related to ethics and compliance policy. As the digital age develops, technology challenges traditional ethical decisions and creates new social issues [71]. This provides insights into companies' efforts to recognize and solve problems.

The sixth topic consists of related words such as business, group, continue, time, innovation, future, increase, gas, importance, data, carbon, cost, project, provide, internal, and governance, esg, and others. It comprises content related to ESG management. ESG is one of the sustainable management strategies of a company that occurs in the risk management dimension and not in the social responsibility dimension [72]. Many companies actively participate in environmental protection activities to reduce greenhouse gas, energy use, and waste emissions. Companies can view sustainable innovation as a strategy for simultaneously pursuing and managing ESG-based social and economic values.

The seventh topic consists of related words such as company, customer, development, service, people, community, digital, support, asset, society, change, increased, healthcare, core, achieve, culture, smart, strategy, position, post, current, target, AI, risk, built, and medical. It can be seen that this comprises content related to digital transformation: the COVID-19 situation further amplified and accelerated the digital transformation of companies [73]. In particular, beyond medical devices and services functioning as one and complementary to each other, software-based medical devices and digital therapeutics are now appearing in the medical field and changing many business models. This indicates that companies contribute to technological development at a rapid pace and formulate various strategies, such as expanding related businesses.

Figures 4 and 5 display the results of Intertopic Distance Map (IDM) analysis for the topics related to COVID-19 extracted through topic modeling. IDM is a diagram that visualizes the size of topics and the distance between them, illustrating the extent to which each topic is related to the others and their degree of similarity. Upon observing the distribution of topics before and after, it is evident that the size of each topic is generally similar with no significant deviation. In addition, the circles do not overlap and are slightly distant from each other, indicating a limited relationship between topics. This suggests that each topic is divided into separate themes [74].

*4.3. Comparative Analysis of Topics before and after COVID-19*

While each topic is distinctly outlined, there are some commonalities or topics that are similar yet different both before COVID-19 and after COVID-19. The initial topics that exhibit consistency in the evolution of corporate management trends are the earlier TOPIC 1 (risk management) and TOPIC 6 (energy innovation), and the subsequent TOPIC 2 (environmental sustainability) and TOPIC 4 (climate change initiatives). Global companies' pursuit of sustainability has been increasingly recognized as an effective strategy for solving a variety of challenges, both before and after COVID-19 [35]. This approach leads

to enhanced competitiveness, improved financial outcomes, and the mitigation of potential business risks [77,78].

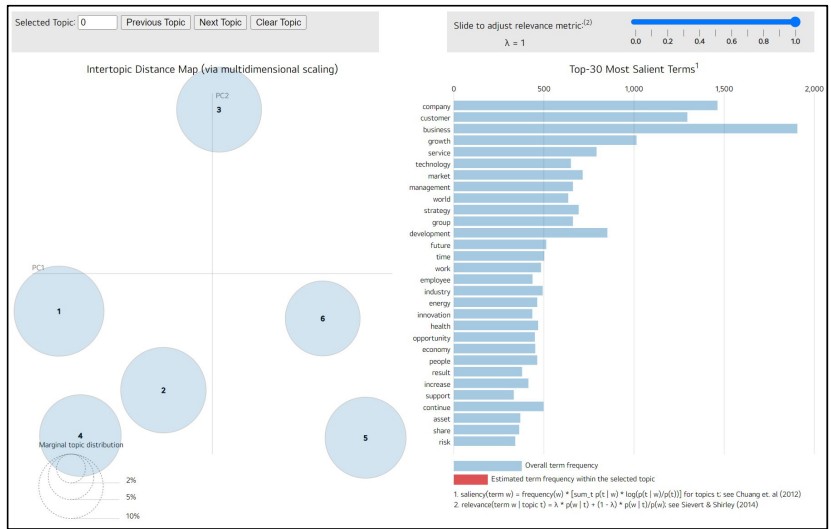

**Figure 4.** IDM results for before COVID-19 [75,76].

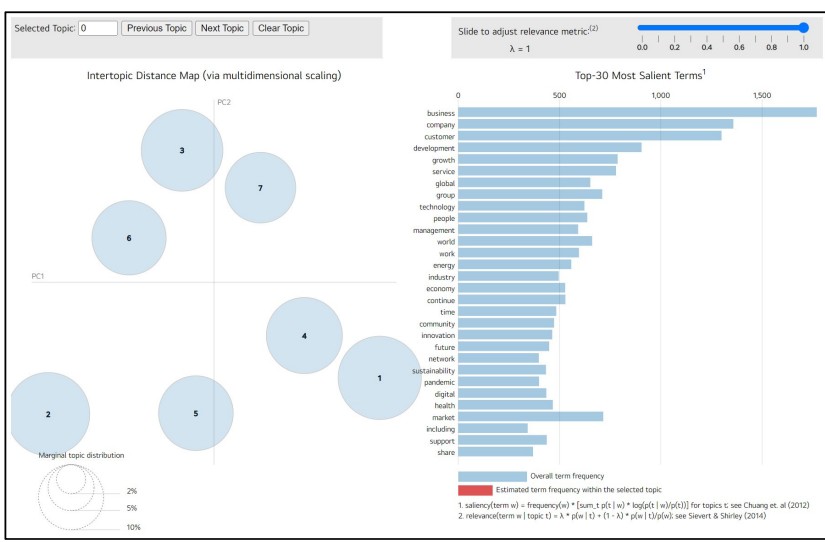

**Figure 5.** IDM results for after COVID-19 [75,76].

The United Nations Global Compact and Business for Scalable Responsibility (2010) report suggests that common sustainability-related risks in many industrial sectors include environmental damage, greenhouse gas emissions, natural disasters, energy consumption, waste, logistics, and transportation. This perspective implies that interest in sustainability stabilized around COVID-19 from the CEO's perspective, after which the significance of climate change, among many other risks, was accentuated. Consequently, issues related to sustainable and environmental innovation strategies, such as carbon neutrality, energy efficiency, and environmentally friendly resources, are gaining prominence.

The second shared theme included TOPIC 2 (market expansion) and TOPIC 3 (operational efficiency), followed by TOPIC 1 (value creation), TOPIC 5 (ethical and compliance policies), and TOPIC 6 (ESG management). ESG issues have garnered attention from businesses and governments since their initial mention in the 2006 United Nations Report [79]. Having evolved into a core component of non-financial outcomes, ESG later developed into the UN Principles for Responsible Investment and expanded into the globally recognized concept of socially responsible investment [80,81]. In major developed countries such as the United States, Europe, and Japan, socially responsible investing, which focuses on

environmentally friendly and ethical companies, has thrived in capital markets. By 2020, the growth rate of ESG funds surpassed USD 1 trillion owing to the impact of COVID-19, leading to a rapid expansion in the scale of global investment associated with corporate ESG activities [80].

ESG issues also impact the operational efficiency of the supply chain, and effective ESG implementation can not only help to reduce increasing operating costs (e.g., raw material costs and actual social costs of water and carbon) but can also significantly impact a company's operating income. Therefore, companies have continued to address a variety of issues in their core strategies, including new value creation and investments, to equip themselves with global competitive advantage.

Although some of these topics are similar, others differ. The first topic with differences in the changing face of corporate strategy trends is the earlier TOPIC 4 (customer-centric innovation), and the later TOPIC 3 (CSR and leadership). This reflects the different types of stakeholders, such as employees, customers, and communities [9]; before COVID-19, customer- and employee-centric marketing strategies were developed to increase customer satisfaction and achieve excellent operational results [82,83].

However, since the outbreak of COVID-19, firms have had to respond quickly to rapid market fluctuations. They also need to improve their overall corporate performance, including economic and social outcomes, to manage risk. Therefore, we identified policy formulation and implementation by the board of directors, including the CEO, as closely related to CSR strategies [84]. Thus, since COVID-19, the topic addressing corporate leadership has been gaining ground as a core strategic element from a CSR perspective.

The second topic with a difference is the earlier TOPIC 5 (digital integration) and the later TOPIC 7 (digital transformation): before COVID-19, IT had developed in the Fourth Industrial Revolution, and the industrial world was evolving into the digital age [59,85]. However, despite technological innovations and various avenues to implement the program in governments, companies, and society as a whole, digital transformation took much longer and faced many difficulties; ultimately, digital transformation issues gradually declined [60,86].

However, after COVID-19, the feasibility of digitalization increased rapidly [87]. A crisis of unprecedented scale demonstrated the impact of digital transformation, as companies responded quickly to change. At Build 2020, Microsoft CEO Satya Nadella said, "COVID is more critical than ever, with digital transformation taking two years to achieve in the past two months" [88]. Thus, companies went from an emergency response to a recovery phase, and the emergence of this issue highlights the importance of digital technology in establishing corporate strategies in the future. Figure 6 illustrates the topics that show similarities and differences as a result of the IDM analysis.

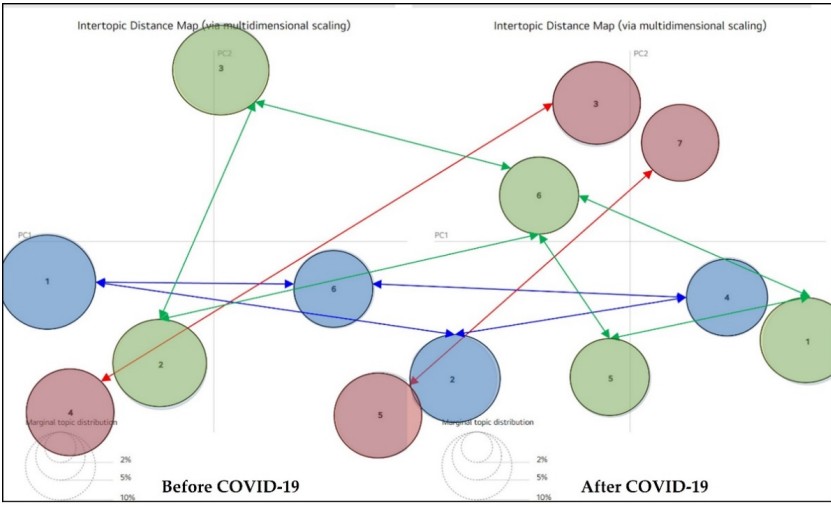

**Figure 6.** Comparison of topics before and after COVID-19.

## 5. Discussion

This study utilizes CEO messages in the CSR reports of the Global Fortune Top 100 companies during the COVID-19 pandemic to perform a comparative analysis of corporate trend changes from the CEO's perspective before and after COVID-19 through simple frequency analysis and LDA topic modeling. Utilizing the Global Fortune company list, which ranks companies by annual sales, allows for the interpretation of global companies with diverse cultural backgrounds and time periods rather than focusing on specific countries or industries. This approach addresses the shortcomings of previous studies, which lacked sufficient data for the pandemic period, thus enabling more accurate value derivation. See the Appendix A for a list of Fortune Global 100 companies. The results of this study are as follows.

A simple frequency analysis by period before and after COVID-19 reveals that during the first period (2017–2019), keywords related to shared culture, such as mutual face-to-face interaction and cooperation, were more heavily weighted, indicating that content related to technological innovation was a priority. In the latter period (2020–2022), keywords related to community-centered content, such as employee health and welfare, the reduction in various inequalities, and sustainable cities were more important, indicating that content related to full-scale digital transformation had become a priority.

Furthermore, by synthesizing the results of the topic-modeling analysis using the LDA technique and the comparative analysis before and after COVID-19, we identified the core keywords of the topics and named them as follows: "risk management", "market expansion", "operational efficiency", "customer-centric innovation", "digital initiative", and "energy innovation" in the first term, and "value creation", "environmental sustainability", "CSR and energy innovation" in the second term. In the latter period, we selected "value creation", "environmental sustainability", "CSR and leadership", and "climate change" as the core keywords. In the latter period, we identified the core keywords as "value creation", "environmental sustainability", "CSR and leadership", "climate change initiatives", "ethical and compliance policies", "ESG management", and "digital transformation". Based on this, we identified commonalities and differences in corporate trends. The first topics with common issues were earlier TOPIC 1 and TOPIC 6 and subsequently TOPIC 2 and TOPIC 4. As environmental awareness has grown worldwide, corporate interest has also increased [10]. The COVID-19 pandemic has led to a greater emphasis on the relationship between social issues and the pursuit of corporate sustainability. The second topic included TOPIC 2, TOPIC 3, TOPIC 1, TOPIC 5, and TOPIC 6. ESG management is an important emerging social issue, both domestically and internationally [80]. As environmental awareness grows worldwide, socially and environmentally sensitive companies such as those in the oil and gas, chemical, automotive, and computer industries are focusing on environmental innovations that require sustainable development [10,80]. This may prompt a paradigm shift toward eco-industries with respect to the risks of climate change and the need for sustainable development.

Looking at the topics that made a difference, the first topic was TOPIC 4 in the previous section and TOPIC 3 in the later section. In the past, the impact on corporate performance was primarily customer-centric. However, the crisis caused by the pandemic changed the way that companies pursue their environmental, social, and economic goals [1]. Companies needed to respond to the pandemic in a way that recognized and addressed the rapidly changing environment and issues in line with the demands of domestic and international customers, investors, suppliers, and various other internal and external stakeholders for CSR activities [4,5]. Thus, the pandemic allowed people to understand that various stakeholders have a significant impact on business operations and the increasingly important role that companies must play in society [1]. The second topic was the previous TOPIC 5 and the later TOPIC 7. Interest in the development of digital technology has amplified during the era of the Fourth Industrial Revolution. However, past studies have shown that up to 70% of digital transformations fail [23,89], because change is difficult to implement on a large scale. However, since then, the entire world has experienced a

crisis of unprecedented scale, which has led to rapidly changing business models [5]. This allows us to understand that companies are trying to gain a competitive advantage in their internationalization strategies by leveraging new digital technologies and business processes for digital transformation [90].

Therefore, a comprehensive analysis of the study results reveals that the primary change after COVID-19, in comparison to before COVID-19, is the prominence of decarbonization as a strategy to address societal environmental problems, such as climate change and global warming. The second change emphasizes digital transformation as a strategy for the new digital age and sustainable management development, owing to increased online activity. Finally, the analysis indicates that CSR activities can be expanded through these core strategies to build trust with diverse stakeholders and anticipate international competitive advantages. To overcome organizational barriers hindering such efforts, strong determination from top management, including CEOs, is crucial above all else [91]. With proactive determination from top executives, even amidst significant changes and crises, organizations can build a differentiated reputation, satisfying the expectations of stakeholders and the general public. Based on this premise, the significance of this study lies in identifying CSR strategies, such as introducing new management approaches and sustainable operational management methods, for businesses to transition their business paradigms during global crises. This can be achieved through messages conveyed by CEOs and executives to various stakeholders domestically and internationally, shedding light on trends in corporate responses to crises.

## 6. Conclusions

The theoretical implications of this study are as follows. The study utilizes CEO message data from CSR reports, which were not often covered in the comparative analysis before and after COVID-19, to understand changes in corporate strategy trends before and after COVID-19. Additionally, while most previous studies were conducted during the pandemic and had limitations in the amount of data [18], this study is significant in that it increases the accuracy of the results by separately analyzing the 2017–2019 period and the 2020–2022 period.

The practical implications of this study are as follows. Firstly, this paper identified themes related to CSR issues and changes before and after COVID-19 based on CEO message data analysis, reflecting CEO perspectives. Second, the results raise awareness of the importance of messages sent by CEOs and boards to various stakeholders through CSR reports during crises at the global level. Third, based on the keyword rankings resulting from this study, we can confirm the factors that companies consider important, regardless of the crisis, thereby providing the information needed for future strategies for CSR reporting by small and medium-sized companies as well as large companies. Additionally, this study compares the CSR-reporting strategies of companies operating in different countries and industries worldwide. Used in various contexts across sectors, this finding suggests that corporate CEOs and boards should pay more attention to the ways in which businesses and society communicate. This implies that rapid decision-making and action by companies in times of crisis should be given the highest priority [2].

Based on the theoretical and practical implications of this study, we conclude that it can serve as theoretical support for the development of global companies' business process strategies and CSR-reporting activities that can effectively communicate with diverse stakeholders during future crises.

However, this study has certain limitations. First, all the CSR reports collected in this study were prepared in English, which excluded CEO messages from some countries and may have resulted in a biased sample. Future research could address this limitation by including multiple languages to elicit a more diverse set of perspectives from different countries. Second, while this study focused on the COVID-19 situation and integrated countries and industries to derive the results, a comparative analysis of the perspectives in the report separately by country and industry would also be valuable. Thirdly, although

there are various methods of text-mining analysis in addition to topic modeling, this study only addressed the LDA topic-modeling method and lacked a multifaceted analysis through other network and sentiment analyses. Therefore, future research that utilizes a more diverse analysis and presents results could provide a more detailed examination of a company's business processes from the CEO's perspective. Lastly, this study extracted data without considering industries, and hence future research could explore how the distribution of industries varies among the top 100 companies in Fortune Global based on revenue. By comparing and analyzing the differences in strategies across industries, insights into future innovation strategies can be provided.

**Author Contributions:** Conceptualization, H.P., T.K. and K.C.; methodology, H.P.; software, H.P.; validation, H.P.; formal analysis, H.P.; writing—original draft preparation, H.P.; writing—review and editing, H.P. and K.C.; supervision, K.C. All authors have read and agreed to the published version of the manuscript.

**Funding:** This research received no external funding.

**Institutional Review Board Statement:** Not applicable.

**Informed Consent Statement:** Not applicable.

**Data Availability Statement:** The data used to support the findings of this study are included in the article.

**Conflicts of Interest:** The authors declare no conflicts of interest.

## Appendix A

**Table A1.** Fortune Global Top 100 Companies List (2022 standard).

| Rank | Company Name |
| --- | --- |
| 1 | Walmart (Bentonville, AR, USA) |
| 2 | Amazon (Seattle, Washington, DC, USA) |
| 3 | State Grid (Beijing, China) |
| 4 | China National Petroleum (Beijing, China) |
| 5 | Sinopec Group (Beijing, China) |
| 6 | Saudi Aramco (Dhahran, Saudi Arabia) |
| 7 | Apple (Cupertino, CA, USA) |
| 8 | Volkswagen (Wolfsburg, Germany) |
| 9 | China State Construction Engineering (Beijing, China) |
| 10 | CVS Health (Woonsocket, RI, USA) |
| 11 | UnitedHealth Group (Minnetonka, MN, USA) |
| 12 | Exxon Mobil (Irving, TX, USA) |
| 13 | Toyota Motor (Toyota City, Aichi, Japan) |
| 14 | Berkshire Hathaway (Omaha, NE, USA) |
| 15 | Shell (The Hague, The Netherlands) |
| 16 | McKesson (Irving, TX, USA) |
| 17 | Alphabet (Mountain View, CA, USA) |
| 18 | Samsung Electronics (Suwon, South Korea) |
| 19 | Trafigura Group (Geneva, Switzerland) |
| 20 | Hon Hai Precision Industry (New Taipei City, Taiwan) |
| 21 | AmerisourceBergen (Chesterbrook, PN, USA) |
| 22 | Industrial & Commercial China (Beijing, China) |
| 23 | Glencore (Baar, Switzerland) |
| 24 | China Construction Bank (Beijing, China) |
| 25 | Ping An Insurance (Shenzhen, China) |
| 26 | Costco Wholesale (Issaquah, Washington, DC, USA) |
| 27 | TotalEnergies (Courbevoie, France) |
| 28 | Agricultural Bank of China (Beijing, China) |
| 29 | Stellantis (Amsterdam, Netherlands) |
| 30 | Cigna (Bloomfield, CT, USA) |

**Table A1.** *Cont.*

| Rank | Company Name |
|------|--------------|
| 31 | Sinochem Holdings (Beijing, China) |
| 32 | AT&T (Dallas, TX, USA) |
| 33 | Microsoft (Redmond, Washington, DC, USA) |
| 34 | China Railway Engineering Group (Beijing, China) |
| 35 | BP (London, UK) |
| 36 | Cardinal Health (Dublin, OH, USA) |
| 37 | Chevron (San Ramon, CA, USA) |
| 38 | Mercedes–Benz Group (Stuttgart, Germany) |
| 39 | China Railway Construction (Beijing, China) |
| 40 | China Life Insurance (Beijing, China) |
| 41 | Mitsubishi (Tokyo, Japan) |
| 42 | Bank of China (Beijing, China) |
| 43 | Home Depot (Atlanta, GA, USA) |
| 44 | China Baowu Steel Group (Shanghai, China) |
| 45 | Walgreens Boots Alliance (Deerfield, IL, USA) |
| 46 | JD.com (Beijing, China) |
| 47 | Allianz (Munich, Germany) |
| 48 | AXA (Paris, France) |
| 49 | Marathon Petroleum (Findlay, OH, USA) |
| 50 | Elevance Health (Houston, TX, USA) |
| 51 | Kroger (Cincinnati, OH, USA) |
| 52 | Gazprom (Moscow, Russia) |
| 53 | Ford Moter (Dearborn, MI, USA) |
| 54 | Verizon Communications (New York City, NY, USA) |
| 55 | Alibaba Group Holding (Hangzhou, China) |
| 56 | Fortum (Espoo, Finland) |
| 57 | China Mobile Communications Group (Beijing, China) |
| 58 | China Minmetals (Beijing, China) |
| 59 | BMW Group (Munich, Germany) |
| 60 | China Communications Construction (Beijing, China) |
| 61 | Honda Motor (Tokyo, Japan) |
| 62 | Deutsche Telekom (Bonn, Germany) |
| 63 | JPMorgan Chase (New York City, NY, USA) |
| 64 | General Motors (Detroit, MI, USA) |
| 65 | China National Offshore (Beijing, China) |
| 66 | Centene (St. Louis, MO, USA) |
| 67 | Lukoil (Moscow, Russia) |
| 68 | SAIC Motor (Shanghai, China) |
| 69 | Shandong Energy Group (Jinan, China) |
| 70 | China Resources (Hong Kong, China) |
| 71 | Meta Platforms (Menlo Park, CA, USA) |
| 72 | Assicurazioni Generali (Trieste, Italy) |
| 73 | Comcast (Philadelphia, PN, USA) |
| 74 | Phillips 66 (Houston, TX, USA) |
| 75 | Hengli Group (Dalian, China) |
| 76 | Amer International Group (Shenzhen, China) |
| 77 | Xiamen C&D (Xiamen, China) |
| 78 | Itochu (Osaka, Japan) |
| 79 | China FAW Group (Changchun, China) |
| 80 | Sinopharm (Beijing, China) |
| 81 | China Post Group (Beijing, China) |
| 82 | Valero Energy (San Antonio, TX, USA) |
| 83 | Nippon Telegraph and Telephone (Tokyo, Japan) |
| 84 | Credit Agricole (Montrouge, France) |
| 85 | China Energy Investment Group (Beijing, China) |
| 86 | Dell Technologies (Round Rock, TX, USA) |
| 87 | Target (Minneapolis, MN, USA) |
| 88 | Mitsui (Tokyo, Japan) |

**Table A1.** *Cont.*

| Rank | Company Name |
|------|--------------|
| 89 | China Southern Power Grid (Guangzhou, China) |
| 90 | Enel (Rome, Italy) |
| 91 | COFCO (Beijing, China) |
| 92 | Hyundai Motor (Seoul, South Korea) |
| 93 | Fannie Mae (Washington, DC, USA) |
| 94 | Japan Post Holdings (Tokyo, Japan) |
| 95 | Electricite de France (Paris, France) |
| 96 | Huawei Investment & Holding (Shenzhen, China) |
| 97 | United Parcel Service (Atlanta, GA, USA) |
| 98 | Life Insurance Corp. of India (Mumbai, India) |
| 99 | Deutsche Post DHL Group (Bonn, Germany) |
| 100 | Power China (Beijing, China) |

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
