# Peer review of "Changes in Management Trends in 100 Global Companies before and after COVID-19: A Topic Modeling Approach"

_sustainability, doi:10.3390/su16062342_

Round 1
Reviewer 1 Report
Comments and Suggestions for Authors
Dear author/s,
Thank you so much for the opportunity to read this very interesting paper.
Good Luck!
Kind Regards,
The Reviewer

Professional proofreading is strongly recommended.
Author Response
Thank you for reviewing our paper, I will submit the revised response incorporating your feedback.
We heartily thank all the comments and coordination again.

Reviewer 2 Report
Comments and Suggestions for Authors
1. Abstract:
- most abstracts are usually around 100–300 words, but yours is 362 words. You should try to shorten your abstract by removing unnecessary details or redundant information and focusing on the main points and contributions of your research.
- You should follow a clear and logical structure, which stands for: Introduction, Methods, Results, and Discussion.
- You should use simple and direct language, avoid jargon and technical terms, and define any acronyms or abbreviations.
- You should also check your grammar, spelling, and punctuation for any errors or typos.
- You should state your research question or problem, your main argument or hypothesis, and your main results and conclusions.
- You should also explain why your research is important, original, and relevant to your field and audience.
- The abstract is not satisfactory as it lacks the key elements that were discussed earlier, such as the practical applications of the study. The summary should be revised to incorporate these elements.
2. Keywords:
- use keywords that reflect the main concepts and terms of your research, and that can help other researchers find your paper or article.
3. Introduction:
- There are some typographical and linguistic errors. The paper must be reviewed, and the errors corrected, or it must be presented to language experts.
- The introduction of the study is unclear and incomplete. It does not state the research aims or the significance of the study. It also does not describe the research population or the sampling method. It only mentions the statistical program that was used for data analysis. Moreover, the introduction does not follow a logical structure, such as presenting the objectives, methodology, research community, scientific contributions, practical applications, and search parameters in a coherent order. The introduction needs to be revised to provide more clarity and information for the readers.
4. literature review
- The literature review lacks a clear introduction that states the research question, objectives, and scope of the review. It also does not provide a thesis statement or a central argument that guides the review. A good introduction should give the reader an overview of the topic, the context, the main issues, and the purpose.
- The literature review does not follow a coherent structure or a logical flow of ideas. It jumps from one subtopic to another without clear transitions or connections. It also mixes different types of literature, such as theoretical, empirical, and methodological, without distinguishing them or explaining their relevance.
- The literature review does not critically evaluate or synthesize the sources. It mostly summarizes and paraphrases the existing literature, without analyzing, comparing, contrasting, or commenting on their strengths, weaknesses, or implications. It also does not identify any gaps or limitations in the literature, or any areas for further research.
- The literature review does not use proper citations and references. It uses a mix of different citation styles, such as APA, MLA, and Chicago, without consistency or accuracy. It also does not provide a complete and alphabetical reference list at the end of the review. Some of the sources are outdated, irrelevant, or unreliable.
5. Methodology:
- The Materials and Methods section contains unnecessary details, such as the name of the journal that publishes the Fortune Global list, the exact dates of the WHO declarations, and the file formats of the reports.
- It also uses vague terms, such as "other reports" and "keywords and topics".
- Please, avoid jargon and technical terms, and define any acronyms or abbreviations
- mixes different types of information, such as the subjects, the data collection period, the data sources, the data extraction, and the data analysis, without distinguishing them or explaining their relevance.
- It does not explain why the Fortune Global 100 largest companies were selected as the subjects, why the data collection period was divided into two phases, how the CEO messages were extracted from the reports, what criteria were used to exclude some companies, what software or tools were used to convert the files, and what methods or techniques were used to analyze the data and identify the keywords and topics. It also does not report any ethical issues, such as how the data were stored and protected, and how the consent of the companies or the CEOs was obtained.
6. conclusion
- The conclusion section is too long and repetitive. It restates the same points that were already made in the introduction and the discussion sections, without adding any new insights or implications. It also uses too many words to convey simple ideas.
- It does not follow a logical order or a consistent format. It mixes different types of information, such as the theoretical implications, the practical implications, the limitations, and the future research directions.
- It contains grammatical, spelling, and punctuation errors, as well as vague, ambiguous, or redundant expressions.
- It does not state the main contribution or significance of the study, or how it addresses the research problem or question. It also does not explain why the findings are important, original, or relevant to the field and the audience. It also does not provide any recommendations, solutions, or actions that can be derived from the study.
7. References:
- In general, there are many old references, some of which can be deleted or replaced with modern references because the topic under study is not considered a precedent and is not characterized by a scarcity of references, studies, and scientific research.
- Your references are mostly relevant to your topic of changes in management trends among global companies before and after the onset of COVID-19. However, some references are not directly related to your topic or are too general. For example, references 10, 11, 12, 15, 16, 17, please up to date the refences and see:
Bello-Pintado, A., & De Massis, A. (2021). COVID-19 and the future of family business research: A literature review and future research agenda. Journal of Family Business Strategy, 12(1), 100421.
https://www.forbes.com/sites/forbesbusinesscouncil/2023/10/26/how-management-has-changed-since-covid-19/
• Cao, X., & Liu, Q. (2020). The impact of COVID-19 on global leadership and cross-cultural management. Frontiers of Business Research in China, 14(1), 1-11.
https://www.cambridge.org/core/journals/journal-of-management-and-organization/article/human-resource-management-and-the-covid19-crisis-implications-challenges-opportunities-and-future-organizational-directions/6857481FD64558659EE4C17C6DAE9AB9
• D’Amato, A., & Zollo, L. (2020). The impact of COVID-19 on corporate social responsibility and marketing philosophy. Journal of Business Research, 116, 176-182.
https://www.hbs.edu/ris/Publication%20Files/20-127_6164cbfd-37a2-489e-8bd2-c252cc7abb87.pdf
Comments on the Quality of English Language
- It contains grammatical, spelling, and punctuation errors, as well as vague, ambiguous, or redundant expressions.
Author Response

(The authors gave the same response as above.)

Reviewer 3 Report
Comments and Suggestions for Authors
Dear authors,
I want to thank you for the opportunity to read your manuscript.
The title got me interested. All who are interested in management, and I think in business in general, should be eager to learn about shifts in “Management Trends.” Of course, the scope, just by reading the title, could be extremely broad or not so, depending on how one would understand the words “Management Trends”. I do not think it’s a weakness (of a title). I think it might evoke the interest of more readers. You scope it around very large firms, which is all good, and then, at least, readers should know what to expect.
But when it comes to the Abstract, you must keep the readers’ attention to get them to read further. I am not sure you managed to do that. I believe all your readers would expect and know already about, what trends and/or themes were the most common ones during Covid-19. Post Covid, it might be less precise. Still, those you mention are of no surprise. To keep readers’ interest, can you flesh out what distinguishes your research findings from the others you have read? I think you need that to get the reader to continue reading. Does your research have a contribution above counting frequencies of words/phrases / topics being addressed? Is there anything you might be able to elaborate on those findings? Your approach is exciting. I think that some might continue reading to learn about your method.
Then, I lose you completely in the Introduction. I see no connection between the Abstract and the Introduction. I thought, when done reading the Abstract, that I would the Introduction be reading about “Management Trends.” Like an introduction to the whole spectrum of the different trends that continue to run through management throughout the decades. Management has gone through a variety of trends during the last 70 years, and it's fascinating to trace them and find arguments for why they appear (e.g., Covid-19, breakthrough research of academics, fieldwork by consultancies, etc.) and how they were applied, with that results, and how they disappeared. We can learn from that. But the introduction suddenly focuses on sustainability, CRS, etc. I admit I don’t follow.
The Lit Rev continues with a sustainability description. Now I think that the abstract is just not intact with the research. When reading the second sub-title of that section, Prior Research on COVID-19 and CEO Messages, I believed I would get back to what the abstract describes and how / what messages are coming from the CEOs of the largest firms. But that’s not the case. So, I'm still a bit lost here. In your Lit Rev, I encourage you to write about how research has managed to read CEO messages, and then simply on sustainability / CSR, if that is your focus (not Management Trends).
When reading the Method, it is there where I believe I read about your intent: “To examine how companies communicated their CSR and sustainability strategies.” So the research is not on management trends, it's on CSR/sustainability. This makes more sense than what one reads after reading the title and the abstract. But then the title and the abstract are very misleading!
What list do you call: “We reviewed the Fortune Global list based on 2022 data.”? Is that the annual report? Or the sustainability reports? Or else?
In your sample, you claim that you have “70 companies, with full texts of 420 CEO messages by fiscal year, which were selected for the final analysis.” Does this mean that you have a 6 messages pr. CEO? It would be interesting to know from what type of reports these might come from and how many reports per company on average there are.
Then, I read your findings. To my surprise, and maybe not, the key words / most common themes are not just on sustainability / CSR. They are various themes from pre-, during, and post-Covid. As expected. Hence, now I do not understand why your Introduction and Literature Review is about sustainability / CSR. This is a major fault in your paper. I think you are not doing your research justice. You have interesting research, but how you write it up, with this ill-linked Intro and Lit Rev, makes no sense.
Please consider rearranging for this or provide an argument for why you do the manuscript as you do it.
I wish you all the best of luck.
Comments on the Quality of English Language
English is fine
Author Response

(The authors gave the same response as above.)

Round 2
Reviewer 2 Report
Comments and Suggestions for Authors
Abstract:
The efforts of the researchers are acknowledged, and it is anticipated that the suggested changes have enhanced the quality and depth of the paper.
Methodology:
adjustments have been made to the methodology section. These changes aim to strengthen the rigor and validity of the research process.
Results:
The findings have been refined based on the recommended revisions.
Discussion:
The discussion section has been updated to incorporate the feedback provided. By addressing the raised points and integrating additional insights.
Conclusion:
In conclusion, the revisions made to the research paper have been guided by the valuable feedback received.
Reviewer 3 Report
Comments and Suggestions for Authors
I am happy to read how the authors have considered the earlier suggestions and comments. The manuscript reads much better and is a very "full" one. I congratulate the authors on a job well done. There are slight issues with the smoothness of English, but those are minor.
Comments on the Quality of English LanguageThe quality of English is good. Today we have so many great tools aiding us when writing English, this should not really be an issue, not any more. There are slight roughness in flow of text but minor.